# Controlling Eutrophication of Aquaculture Production Water Using Biochar: Correlation of Molecular Composition with Adsorption Characteristics as Revealed by FT-ICR Mass Spectrometry

William F. Rance Bare [1], Ethan Struhs [2], Amin Mirkouei [1,3,*], Kenneth Overturf [4], Martha L. Chacón-Patiño [5,6], Amy M. McKenna [4,7], Huan Chen [5] and Krishnan S. Raja [3]

[1]  Department of Biological Engineering, University of Idaho, Idaho Falls, ID 83402, USA;
    bare9199@vandals.uidaho.edu
[2]  Department of Mechanical Engineering, University of Idaho, Idaho Falls, ID 83402, USA;
    stru4589@vandals.uidaho.edu
[3]  Department of Nuclear Engineering and Industrial Management, University of Idaho,
    Idaho Falls, ID 83402, USA; ksraja@uidaho.edu
[4]  Agricultural Research Service, United States Department of Agriculture, Hagerman, ID 83332, USA;
    ken.overturf@usda.gov (K.O.); mckenna@magnet.fsu.edu (A.M.M.)
[5]  National High Magnetic Field Laboratory, Ion Cyclotron Resonance Facility, Florida State University,
    Tallahassee, FL 32310, USA; chacon@magnet.fsu.edu (M.L.C.-P.); huan.chen@magnet.fsu.edu (H.C.)
[6]  International Joint Laboratory for Complex Matrices Molecular Characterization, iC2MC, TRTG,
    76700 Harfleur, France
[7]  Department of Soil and Crop Sciences, Colorado State University, Fort Collins, CO 80523, USA
*   Correspondence: amirkouei@uidaho.edu

**Abstract:** This study aims to construct a novel and sustainable approach for remediating aquaculture-generated water contamination using various engineered biochars. Particularly, this study focuses on capturing nitrogen and phosphorus from downstream water of commercial fish farms in Magic Valley, Idaho, containing approximately 2.26 mg/L of nitrogen and 0.15 mg/L of phosphorous. The results indicate that the proposed approach can improve downstream waters by adsorbing micronutrients (e.g., nitrogen-ammonia, nitrate-n + nitrite-n, and total phosphorus). Water treatment time and biochar pH are two key parameters strongly associated with adsorbed compounds. Molecular-level characterization of solvent-extracted organics from biochar materials (before and after water treatment) suggests increased levels of highly oxygenated molecules as a function of increasing water treatment time. Also, the results show the enrichment in organic species with higher molecular weight and increased double bond equivalents, with a compositional range similar to that of dissolved organic matter. Upon water treatment, extracted organics revealed higher abundances of compounds with higher H/C and O/C ratios. The engineered biochars, after water treatment, can be reused as nutrient-rich fertilizers. This study concluded that the engineered biochars could sequester more nitrogen and phosphorous over time. Also, the proposed approach can simultaneously increase fish production capacity and support the aquaculture industry in different regions by improving water quality and enabling aquaculture expansion.

**Keywords:** aquaculture production; engineered biochar; eutrophication; FT-ICR mass spectrometry; water treatment

## 1. Introduction

### 1.1. Challenges and Needs

The aquaculture industry produces aquatic organisms and seafood (e.g., fish, shellfish, and crustaceans) that can help to meet the growing consumer demand for healthy protein using natural, domestic resources [1]. According to the National Oceanic and Atmospheric

Administration (NOAA), the global seafood market reached $406 billion, and the United States (U.S.) imported over $21 billion (over six billion pounds) in seafood in 2020 [2]. Therefore, a sustainable aquaculture industry can provide an affordable and safe supply of seafood products, as well as train the workforce and create jobs in rural communities. However, the aquaculture industry is one of the prominent consumers of global water resources and one of the leading causes of eutrophication and water degradation [3].

Eutrophication is the result of oversupplied nutrients (e.g., nitrogen and phosphorus), pollutants, and wastes (e.g., uneaten fish food, fish feces, and drugs) from various operations (e.g., agriculture and aquaculture) to water bodies. Eutrophication changes water pH, oxygen, and temperature, as well as reduces water quality due to structural changes. For example, eutrophication can increase algae and marine plants' growth, which can threaten the survival of native fish species. Currently, there is no affordable method for addressing eutrophication and water degradation in large water bodies, such as rivers.

Aquaculture is a critical industry to the state of Idaho in the U.S., with most of the fish production located in southern Idaho (Hagerman area), approximately 40 km (around 25 miles) west of Twin Falls. Idaho permits around 65 aquaculture operations in the Hagerman area to discharge their downstream water into the Snake River. Currently, these fish farms produce 70–75% of all rainbow trout (over 30,000 tons annually) that are grown in the U.S. with over $100 million in annual revenue that has a huge effect on the southern Idaho economy and the employment of almost 1000 low skilled workers, directly or indirectly [4]. However, the U.S. Environmental Protection Agency (EPA) has determined that the discharge of water from aquaculture operations into the mid-reach of the Snake River increases the eutrophication risks, especially dense algal blooms and low water oxygen content that can cause changes to the ecosystem. Therefore, a cost-effective, environmentally friendly method to remove nutrients and pollutants prior to discharge to water bodies can address these challenges and consequently allow for increased production, reduced eutrophication, and economic stability in the region.

### 1.2. Background

Biochar Use for Water Remediation. Biochar has the potential to effectively improve eutrophic water by adsorbing micronutrients and contaminants due to its physicochemical properties. Heating organic materials or wastes (e.g., biomass feedstocks) in the absence of oxygen can thermally decompose their compounds to solid materials (biochar), liquid materials (bio-oil), and gases. These biomaterials have applications in various domains, such as addressing sustainability challenges across food-energy-water systems [5]. Pyrolysis is the main technology for producing biochar from organic materials using high-temperature processes without the presence of oxygen. Earlier studies showed that biochar (similar to charcoal) has the potential to remove micronutrients and pollutants (e.g., arsenic, cadmium, ammonium, phosphorus, and nitrate) from water and effluent streams [6,7]. Biochar is an adsorbent that can hold adsorbates (tiny particles, pollutants, and organic substances) in water due to its unique properties (e.g., surface area, architecture, porosity, and textual features). Recent studies investigated biochar adsorption capacities for immobilizing contaminants and mitigating water degradation [8]. For example, Yin et al. (2018) explored the application of modified biochar to remove ammonium, nitrate, and phosphate from eutrophic water [7]. They used modified biochar and Al-modified biochar from soybean straw, and their results show that Mg-/Al-modified biochar is an excellent material for eutrophic water treatment [7]. Novais et al. (2018) also investigated modified biochar from sugarcane straw and poultry manure for removing phosphorus from eutrophic water [9]. They concluded that biochar could be a safe solution for adsorbing phosphorus from water and reusing it in agriculture practices [9]. Shao et al. (2020) investigated nutrient removal and bacterial immobilization from water bodies in recirculating aquaculture systems using biochar from maize straw [10]. Their results show that biochar can improve the water quality and crab production yield [10]. Lin et al. (2021) applied biochar from spent mushroom substrate to improve water quality in aquaculture ponds, particularly in red claw

crayfish farms [11]. Their results show that spent mushroom biochar can increase the water pH and reduce ammonia-N and sulfite levels [11]. Silva et al. (2021) explored biochar from waste-based materials for removing antibiotics from aquaculture effluents [12]. The latest studies by Chen et al. (2022) investigated biochar to immobilize heavy metals and polycyclic aromatic hydrocarbons in pond sediment and aquaculture products [13]. They concluded that biochar could remediate pollution in aquaculture environments while safeguarding the quality of aquatic products [13]. Also, Hu et al. (2022) and Hung et al. (2022) explored the use of biochar for removing contaminants (e.g., drugs) from aquaculture water bodies and ponds [14,15].

Inherent Organic Compounds in Biochar. Fast pyrolysis of biomass produces carbon residues, biochar, whose composition is shaped by the sequestration of recondensed pyrolysis liquids, consisting of ultra-complex mixtures of organic compounds, predominantly with high oxygen content. Previous reports have used solvent extractions to isolate organic species from biochar and characterize them via Fourier transform ion cyclotron resonance mass spectrometry (FT-ICR MS) [16], which offers the highest resolving power and mass accuracy among all available MS techniques [17,18]. Analysis of organic extracts from biochar has revealed thousands of compositions. In a typical FT-ICR mass spectrum, with tens of thousands of ions, each peak gets assigned a unique molecular formula based on mass accuracy and isotopic composition. Molecular formulae are then sorted into compound classes. For example, compounds with only C and H make up the HC class, whereas species with two N atoms and five O atoms, in addition to C and H, comprise the $N_2O_5$ class. Graphical representation of the molecular composition can be visualized by plotting three-dimensional plots of double bond equivalents (DBE), the number of rings plus double bonds to carbon, versus carbon number, with a color scale highlighting the relative abundance contributions of each compound within that heteroatom class. In addition, van Krevelen diagrams plot atomic H/C vs. O/C ratios [19,20]. Previously, McKenna et al. (2021) used solvent fractionation to extract organics from biochar, followed by subsequent characterization by atmospheric pressure photoionization coupled with 21 T FT-ICR MS [16] and reported functionalized and nonfunctionalized polycyclic aromatic hydrocarbons with aromatic structural motifs [16,21].

### 1.3. Study Focus and Objectives

The main objective of this study is to explore the remediation of aquaculture-generated water by a novel water treatment approach based on engineered biochars from woody feedstocks that have the potential to be recycled and reused as nutrient-rich fertilizers. Our earlier studies show that pinewood can effectively adsorb micronutrients (P and N) from water [22]. Thus, this study focuses on biochars from different pinewoods (e.g., lodgepole or mixed ponderosa pine and spruce trees). The novelty of this study lies in addressing eutrophication risks, using a sustainable method to remove contaminants from aquaculture effluents before discharging them to water bodies. The applied approach could promote sustainable aquaculture practices and create an entirely new industry, employing used biochar after water treatment as nutrient-rich soil conditioners or fertilizers due to its high carbon and nutrient content. We seek to address several research questions related to the adsorption capacity of engineered biochars (especially N and P), the chemistry of adsorbed and released materials after water treatment, and the overall environmental performance of the designed water treatment method. Also, this study explores solutions to address some of the issues associated with the complex molecular nature of biochar filters and soluble organic species that can harbor in their intricate microstructure. Characterization studies via ultra-high-resolution mass spectrometry can provide valuable information on extractable organics and components at the molecular level before and after using biochar for water treatment. The investigation of such a complex mixture is central for potential applications in water treatment. The overall characterization of the materials at the molecular level will enable more accurate modeling and scale-up of valorization

processes that will, in turn, allow more efficient utilization of water and food resources, increasing their economic value and helping the nation's natural supplies.

## 2. Materials and Methods

### 2.1. Materials

Biochar. This study uses two different biochars: (1) from older growth ponderosa pine and spruce trees, with little to no lodgepole in them, and from the slash piles of timbering operations in Medford, Oregon, labeled as OB (Oregon Biochar), and (2) from lodgepole pine in southeast Idaho, labeled as IB (Idaho Biochar). Biochar production processes are slightly different for Oregon and Idaho Biochars; all samples were dry sieved for plus 100 microns before water treatment.

Water. One of the prime prerequisites for establishing an aquaculture operation is a representative water source. In this study, the water is provided from Thousand Springs (the outlet for the east Snake Plain aquifer), which flows more than 5.66 m$^3$/s (around 200 cubic feet per second) from several springs in the canyon wall. The Thousand Springs water is ideal for growing rainbow trout due to the temperature range of 13.5–15.5 °C (57–60 °F) and its quality, such as pH range (Figure 1a).

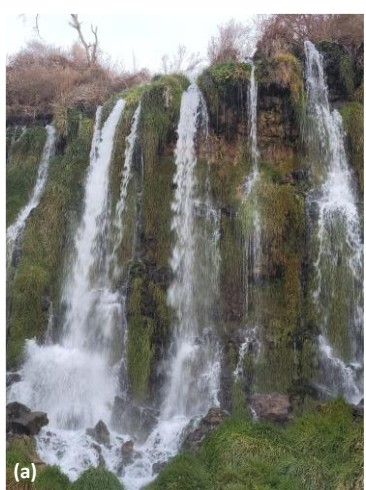 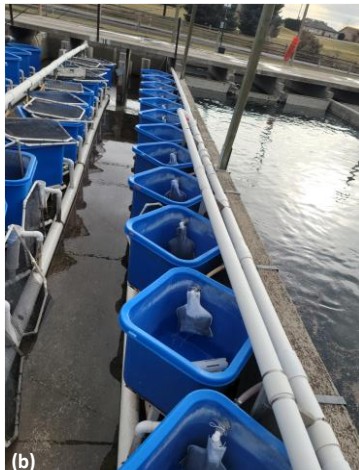 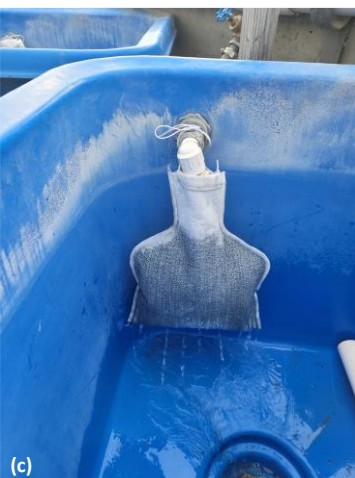

**Figure 1.** (**a**) Thousand Springs water for growing rainbow trout, (**b**) testing tanks, utilizing fourth-use waters from the main rearing ponds that contain relatively higher levels of nitrogen and phosphorus, and (**c**) large size water filter bag with 300 g biochar for 30 min treatment and 450 g biochar for 12–48 h treatment, attached to the water pipe with ties.

### 2.2. Methods

Biochar Production Method. Biochar samples are produced from woody biomass feedstocks (e.g., pinewood) under different process configurations (e.g., conversion temperature and residence time). The IB production process includes size reduction (grinding), drying, and slow pyrolysis (at 400–450 °C under 10–15 psi for 30 min) using the built-in-house slow pyrolysis reactors [23]. The OB was provided by Oregon Biochar Solutions [24].

Water Treatment Method. Field tests were conducted at the USDA facility equipped with water from production fish farms, located north of Buhl, Idaho, adjacent to the Snake River, utilizing fourth-use waters from their main rearing ponds, known to contain relatively higher levels of nitrogen and phosphorus (Figure 1b). The utilized field test area has been developed by the U.S. Department of Agriculture (USDA) in cooperation with the University of Idaho aquaculture research facilities in Hagerman for many field studies and research projects. For the field experiments, each testing tank can control the water flow into the tank. Flow rates for this study were set at 7.57 L/min. Time intervals were designed to establish the saturation time for different sample sizes. Particularly, 24 testing tanks were used, including six experiments (3 for OB and 3 for IB) using 300 g biochar for 30 min;

and 18 experiments (9 for OB and 9 for IB) using 450 g biochar for 12, 24, and 48 h, each time-period in triplicate. Each biochar sample was placed in a large water filter bag, and the bag was attached to the water pipe with zip ties (Figure 1c). The used water filter bags have a pore size of 100 microns. Water treatment experiments were run in triplicate to evaluate the removal efficiency of nitrogen and phosphorus from the aquaculture production water using different biochar samples.

Characterization Methods. Several analytical laboratories assisted this study by analyzing biochar and water before and after treatment. For biochar nitrogen-ammonium analysis, a KCI extractable (ASA 33-3.2) was used for preparation, followed by the colorimetric method (ASA 33-7.3) [25,26]. Similarly, for biochar nitrogen-nitrate + nitrate analysis, a KCI extractable (ASA 33-3.2) was used for preparation, followed by the colorimetric method (ASA 33-8.3) [25,26]. For total phosphorus, biochar was digested in 30% nitric acid and analyzed by Inductively Coupled Plasma Optical Emission Spectrometry (ICP-OES). For water analysis, EPA protocols 365.1 and 300.1 were used to assess total phosphorus and nitrogen, respectively [27,28]. For characterization of the extractable organic compounds, 30 g of biochar were placed in cellulose thimbles and extracted in a Soxhlet apparatus with separate solvent mixtures during 24 h of acetone/cyclohexane 1/1 and toluene/tetrahydrofuran/methanol 1/1/1. The extracted species in solution were desolvated under nitrogen, weighed, and stored in the dark prior to FT-ICR MS analysis.

Molecular Characterization by 21 T FT-ICR MS. 21-tesla FT-ICR MS coupled with negative-ion electrospray ionization (-ESI) was used to (a) compare the extractable organics from different biochar samples before and after water treatment, (b) to understand the effect of biochar production and water treatment conditions, and (c) to identify biochar production conditions that produce materials with high efficiency for the removal of specific water contaminants (e.g., nitrogen and phosphorus containing compounds).

Extracted organics were ionized by negative-ion electrospray ionization, extensively used to characterize polar, oxygen-containing compounds through deprotonation that occurs in solution [29,30]. Fractions were dissolved in 1:1:1 toluene:tetrahydrofuran: methanol at a 150 μg/mL concentration, infused at 0.5 μL/min, and ionized with a needle voltage of −3.0 kV, with typical source conditions, 3. Samples were diluted in Ions and analyzed with a custom-built 21T FT-ICR mass spectrometer, in which $1 \times 10^6$ charges were accumulated for 1–3 ms in an external ion trap using automatic gain control (AGC) [31,32]. Ions were transferred to the dynamically harmonized ICR via a decreasing auxiliary radio frequency. Excitation and detection were achieved on the same pair of electrodes of the ICR cell, operated at 6 V. 100 time-domain transients of 3.1 s were collected and averaged with Predator Software Corev1.2.3 for each sample. Mass spectra were internally calibrated with abundant homologous series based on the "walking" calibration method [33]. PetroOrg© Software NS 18.0.6 [34] assigned elemental composition assignment and data visualization (e.g., DBE vs. carbon number and van Krevelen diagrams). Only classes with a combined %R.A. above 0.15% total RA will be considered.

## 3. Results and Discussion

### 3.1. Results

Biochar Analysis. Both IB and OB experiments appeared to capture (sequester) nitrogen-ammonia, nitrate-n + nitrite-n, and total phosphorus over time. IB experiments show significant nutrient (N and P) adsorption, especially nitrogen-ammonia and nitrate-n + nitrite-n. The results show that IB phosphorus increased from an average of ~240, 380, to 430 μg/g after 12, 24, and 48 h water treatment, respectively (Figure 2a). IB and OB initially released phosphorous into the water but gradually sequestered phosphorous over time. Also, IB and OB samples adsorbed micronutrients from downstream water over time but never sequestered phosphorous within the time frame of the experiments (0.5–48 h) at a level higher than the unused (before treatment) biochars. Figure 2 provides insight into relationships between the experimental parameters of interest. Nitrate, nitrite, and ammonium adsorption efficiency of biochar increased with longer residence times for

IB. Contrary to our previous results of short water treatment (residence) time experiments (5–60 min) for the static water treatment system, there is a visible correlation between the adsorption of phosphorus and nitrogen compounds seen in IB samples [22]. Key parameters (e.g., residence time and biochar pH) had relatively strong associations with adsorbed compounds. OB, compared to IB, has high total phosphorus and pH, which seems to have led to the desorption of phosphorus and a significant decrease in pH with prolonged residence times. Overall, IB results appear more promising as a solution to reduce eutrophication (oversupplied nutrients) from fish farm effluent. Longer residence times should be applied to identify the optimal treatment time for the adsorption of phosphorus and nitrogen compounds. Table S1 in Supplementary Materials provides IB and OB results before and after water treatment.

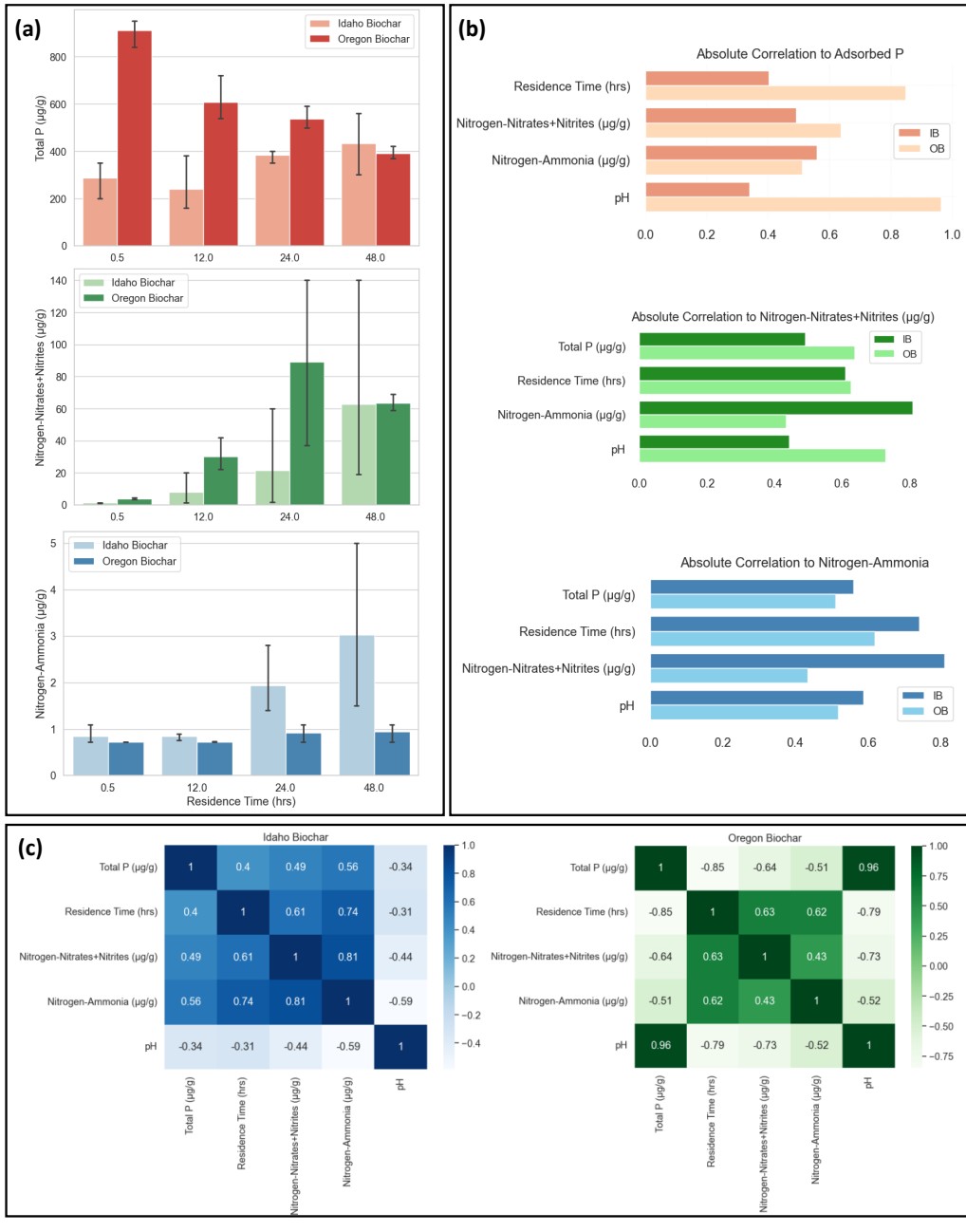

**Figure 2.** Field results and comparison: (**a**) biochar adsorption of phosphorus, nitrates, nitrites, and ammonia after specified residence times; (**b**) absolute Pearson coefficients relative to adsorbed compound; and (**c**) correlation of key experimental parameters by biochar source.

Water Results. During the water tests, water was run through 300 g of biochar for approximately 0.5 h; both IB and OB released phosphorous into the water, but the amount was significantly larger for OB experiments. The water running through the testing tanks used in this study contained approximately 0.15 mg/L of total phosphorous and 2.26 mg/L of nitrate. Table S2 in Supplementary Materials presents water results before and after 0.5 h water treatment, using IB and OB.

Molecular Characterization Results. Biochars, before and after being used in water treatment, were extracted with two separate solvent mixtures. First, acetone/cyclohexane extracts non-polar/moderately polar species entrained within the char complex microstructure. Next, acetone and cyclohexane target the extraction of less polar/highly aromatic compounds [35,36]. The second solvent mixture, toluene/tetrahydrofuran/methanol, assisted in the isolation of more aromatic/polarizable compounds. Each fraction was quantified gravimetrically. Table 1 shows the mass yields and indicates the IB leached higher amounts of organics (around 5-fold) compared to OB, and in general, more material was extracted with toluene/THF/MeOH, suggesting the dominance of polar organics within IB's structure.

**Table 1.** Weight (wt.%) percentage of extracted organics relative to the amount of biochar used in each extraction.

| Sample | Acetone/Cyclohexane | Tol/THF/MeOH | Treatment (h) |
|---|---|---|---|
| Untreated Idaho | 1.9 | 2.8 | 0 |
| IB 1 | 2.4 | 3.5 | 12 |
| IB 2 | 3.1 | 4.6 | 24 |
| IB 3 | 3.6 | 4.7 | 48 |
| Untreated Oregon | 0.1 | 0.3 | 0 |
| OB 1 | 0.1 | 0.6 | 12 |
| OB 2 | 0.4 | 0.7 | 24 |
| OB 3 | 0.3 | 0.9 | 48 |

The separate fractions were analyzed via negative-ion electrospray ionization (-ESI) coupled with 21-tesla FT-ICR MS. Solvent fractionation with different solvent chemistry extends the compositional space for detected species in complex mixtures via mass spectrometry, as previously shown for fossil fuels [17,37]. Tens of thousands of peaks were assigned in each sample based on mass accuracy and isotopic structure. Then, the elemental compositions were sorted based on the number and type of heteroatom (i.e., nitrogen, oxygen, sulfur) classes. Figure 3 presents the heteroatom class distribution for $O_x$ classes detected in the acetone/cyclohexane and toluene/THF/MeOH fractions for IB and OB biochars. Species that contain one oxygen correspond to $O_1$ (phenol, alcohol) in negative-ESI and $O_2$ (carboxylic acid), whereas polyfunctional oxygen moieties comprise classes with more than two oxygens ($O_{3+}$ per molecule). The acetone fraction from IB revealed an increase in oxygen numbers as a function of increasing water treatment time. For instance, the untreated biochar featured abundant compounds with six or fewer oxygens and dominance of $O_4$-$O_6$ classes. Conversely, compounds were detected after water treatment with much higher oxygen numbers (up to $O_{17}$). This effect is less pronounced in the Tol/THF/MeOH extracted organics. Still, for untreated biochars, low-order oxygen-containing classes, such as $O_2$ and $O_3$, are more abundant compared to the extracted organics after water treatment. Compositional trends in the class distribution are more difficult to capture for OB. However, the bar graphs indicate that OB2, after 24 h of water treatment, contains abundant compounds with ultra-high oxygen content (e.g., $O_7$, $O_8$, and $O_9$ classes).

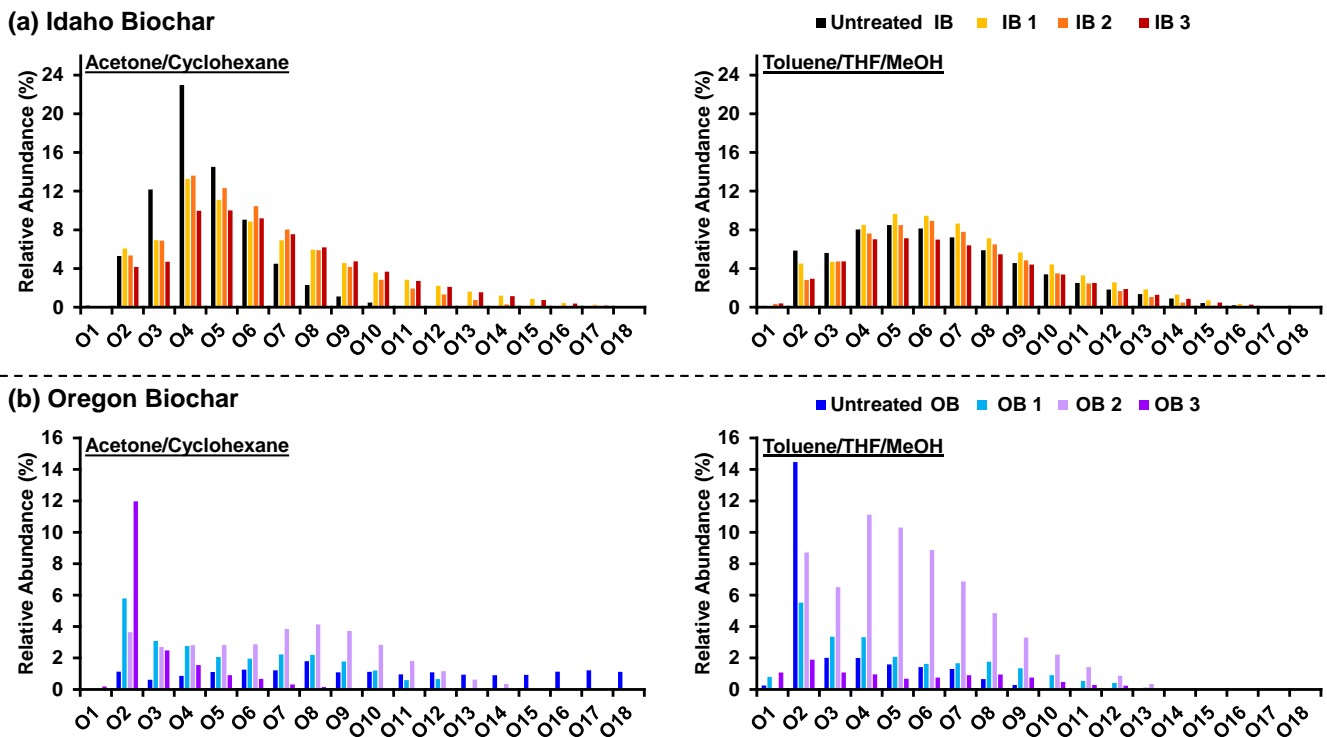

**Figure 3.** Heteroatom class distribution for oxygen classes derived from negative-ion FT ICR Ms at 21tesla for biochar solvent fractions. Ox (O$_1$, O$_2$, ..., O$_{18}$) were identified for the acetone/cyclohexane (left) and toluene/THF/MeOH (right) fractions for (**a**) IB and (**b**) OB samples.

Molecular formulae are represented in compositional plots of DBE vs. carbon numbers for acetone/cyclohexane and Tol/THF/MeOH fractions (Figure 4). The extracted organics from OB feature a more limited compositional range, in terms of both DBE and carbon numbers, with abundant detection of compounds with DBE < 20, afar from the red dotted line. This line is known as the polycyclic aromatic hydrocarbon limit and is a compositional boundary for planar organic molecules. Compositions close to the polycyclic aromatic hydrocarbon limit are highly aromatic/pericondensed and alkyl depleted. The results show that IB samples feature more species with compositions close to the polycyclic aromatic hydrocarbon line, with DBE up to 30. This suggests abundant content of highly aromatic, oxygen-containing molecules, likely similar to oxyPAHs [38], which, to some extent, resemble the composition of natural organic matter [39]. The compositional range for the extracted organics with Tol/THF/MeOH is much more extended and is highly enriched in compositions with DBE values above 20, in which relative abundance increases as a function of increasing water treatment.

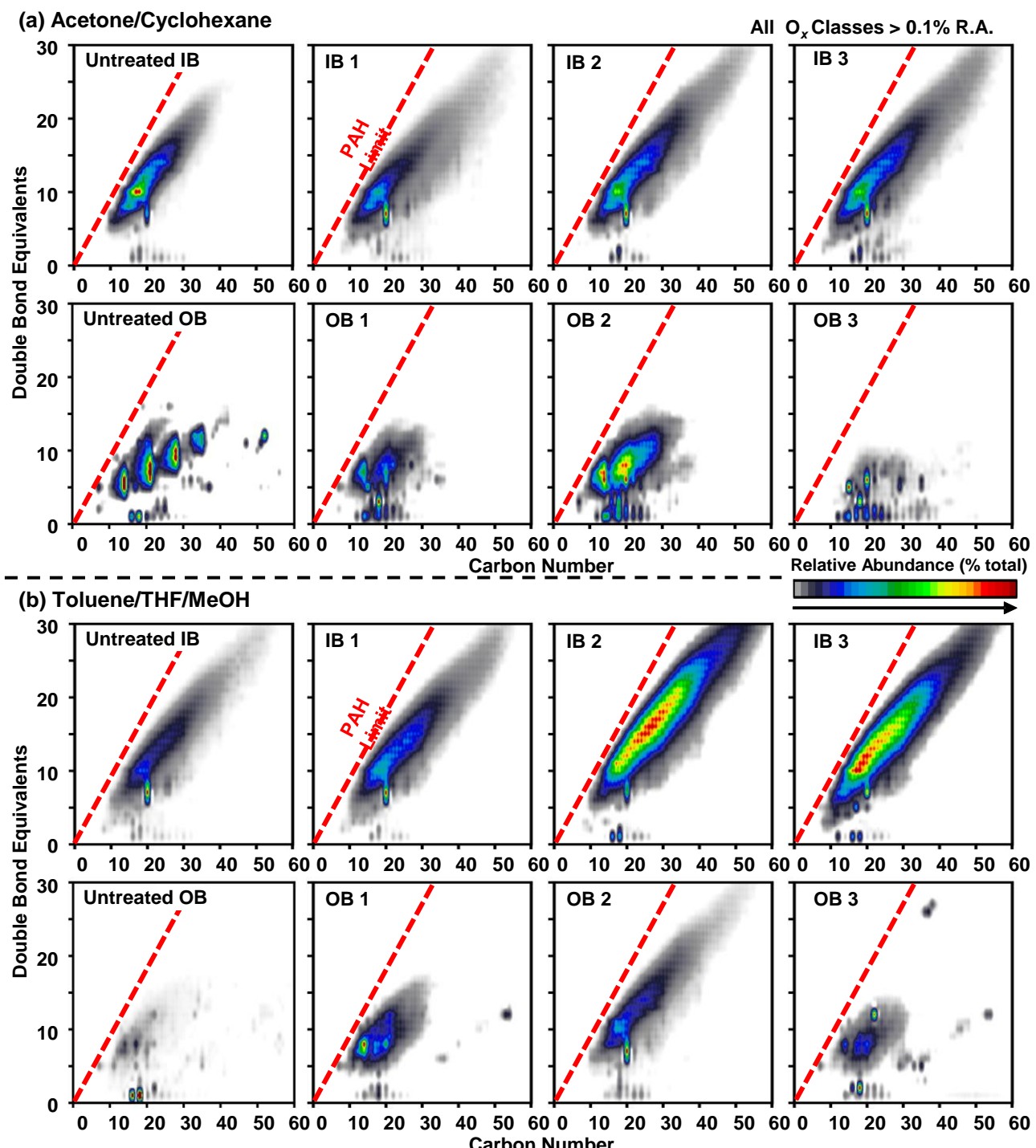

**Figure 4.** Compositional ranges of DBE vs. carbon numbers for (**a**) acetone/cyclohexane and (**b**) Tol/THF/MeOH fractions.

Figure 5 presents the compositional plots of H/C vs. O/C ratios via van Krevelen diagrams. The color scale represents the aromaticity index: non-aromatic (black, AI ≤ 0.5), aromatic (gold, 0.67 > AI > 0.5), and condensed aromatic (red, AI ≥ 0.6 [40]). The results indicate that after 48 h of water treatment, the amount of compositions (each data point contains at least one molecular formula) increases in each of the "aromaticity zones." In other words, after water treatment, the plots contain more data points. Particularly, the acetone fraction for IB reveals dramatic changes in composition.

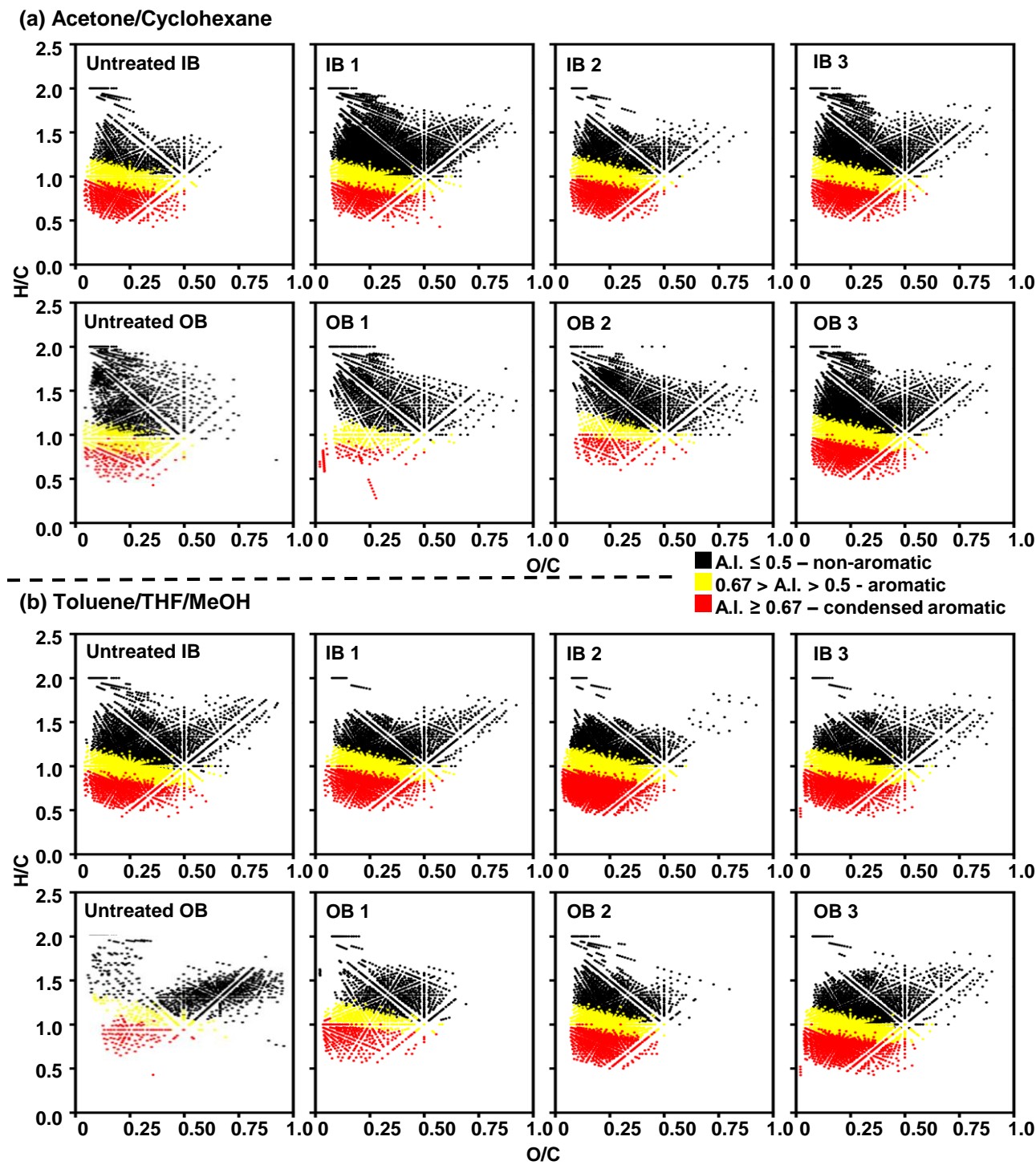

**Figure 5.** Compositional ranges of H/C vs. O/C ratios via van Krevelen diagrams for (**a**) acetone/cyclohexane and (**b**) Tol/THF/MeOH fractions.

Figure 6 presents expanded van Krevelen diagrams for the acetone fractions for IB before and after 48 h of water treatment. The color scale is the relative abundance as detected by MS and normalized within all the plotted compositions. "New" compositions are detected after 48 h of water treatment, as highlighted in red. Such species are likely captured from the water, over time, feature much higher oxygen content (O/C > 0.5), and are likely water soluble.

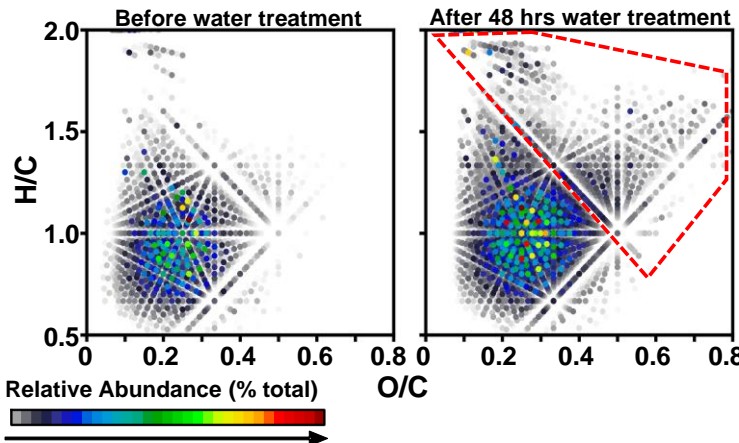

**Figure 6.** Expanded van Krevelen diagrams of acetone/cyclohexane fractions for IB before and after 48 h of water treatment. New compositions are detected after 48 h of water treatment, as shown in the red dashed box.

### 3.2. Discussion

The seafood market has become well-established over the past 40 years and contributes to addressing food supply and local job creation. However, the effluents from the downstream water of these farms include a wide array of contaminants, which result in eutrophication. Earlier studies in 2009 reported that eutrophication cost in the U.S. due to various reasons, such as global warming and water pollution from excess nutrients, is around $2.2 billion annually [41]. Western States (e.g., Idaho and California) have strong aquaculture industries, and the current limitation for fish farms in Idaho for releasing total phosphorus has been set at 0.075 mg/L. These permits were issued in 2012 (or later) and are currently under review for reissue due to high eutrophication risks. Lowering the nutrient limits by EPA and the Idaho Department of Environmental Quality (DEQ) could have devastating effects on the current operations, such as either closing fish farms or reducing their production rate, which can subsequently create significant economic hardship for the entire region.

Sustainable solutions using renewable materials (e.g., biochar) can improve water quality and the performance of the aquaculture industry. The renewable materials sector has frequently been seeking in-depth knowledge about the nature and intrinsic properties of biomaterials. The challenges in resolving the complex composition of thermochemically-derived biochar from biomass feedstocks has precluded its appropriate utilization in renewable applications, leaving open new opportunities for the development of alternative usages, such as water treatment and soil-plant health improvement.

The best result was observed with IB 3.1 (48 h treatment time), as presented in Table S1 (in Supplementary Materials), where the total nitrogen sequestration was 155 μg/g, and phosphorus sequestration was about 60 μg/g. The best result among the OB samples was OB2.2 (24 h treatment time) with a total nitrogen sequestration of 140.93 μg/g. The OB showed no capacity for phosphorus sequestration. In fact, the OB had a higher initial P concentration of 1200 μg/g, and the total P content decreased with the water treatment. The adsorption of nitrogen and phosphorus species by biochar can be better understood by correlating their molecular structures with the adsorption capacity. The adsorption capacity is attributed to chemisorption via electron donor–acceptor (π–π) interactions, interaction with O-containing functional groups forming hydrogen bonds, weaker physisorption (due to van der Waals forces), and ion exchange [42]. Biochar can adsorb N and P simultaneously. However, the inhibiting effect of adsorption among similarly charged anions has been reported due to competition for adsorption sites between $NO_3^-$ and $PO_4^{3-}$ and competition for exchange reactions [43]. The P removal capacity of the biochar is high when the surface is modified by ions of alkali metals, such as Ca and Mg [44]. The lower phosphorus

sequestration levels of OB and IB could be attributed to the absence of alkali metal or Al and Fe cations in the biochar.

The degree of aromatic compounds present in biochar is influenced by the pyrolysis condition. Increasing the temperature, resident time, and air level during pyrolysis would result in a higher level of aromatic compounds in biochar [45]. Very high pyrolysis temperature results in the loss of a large number of polar functional groups, such as -CH, C=C, and C=O, resulting in enhanced hydrophobicity of the biochar surface and poor adsorption characteristics. Slow pyrolyzed biochar reveals predominantly fused aromatic C rings with H bonds due to surface carboxylate or phenolate groups [46]. An increase in the surface negative charge by substituting the edge of the C-H fraction of aromatic rings with $COO^-$ groups under oxidation conditions increases the cation exchange capacity of the biochar. This helps with the adsorption of $NH_4^+$ but not with anions, such as $NO_3^-$ or $PO_4^{3-}$.

The point of zero charge of the pine-derived biochar was reported to be at pH 8.47 [47]. In this study, the pH of untreated water was in the range of 7.61–7.67, which kept the biochar surface in an acidic condition and promoted the adsorption of anions, such as $NO_3^-$, $NO_2^-$, $H_2PO_4^-$, $HPO_4^{2-}$ and $PO_4^{3-}$. It is also noted that the pH varied after the water treatment. The change in the pH ($\Delta$pH) was in the range of $-0.7$ to 3.1, depending on the experimental conditions. A pH condition higher than 8.5 would render the biochar surface negatively charged and, therefore, limit the adsorption of anions. At higher pH conditions, the adsorption of nitrogen in the form of $NH_4^+$ could be promoted.

In this investigation, no intentional metal functional groups were introduced. Both the untreated IB and OB had a nitrogen content of <1.72 μg/g in the starting material, which indicated that both the biochar materials had very limited nitrogen functional groups for effective adsorption. Therefore, the non-metal functional group initially available for adsorption reaction could be the oxygen functional group, such as—COOH, -C=O, -COC-, -COO, -CHO, and -OH [48]. However, after exposure to nitrogen-rich water, the surface of the biochar could be modified with nitrogen species, and nitrogen functional groups could be present in the form of $NH_4$-N, $NO_2$-N, and $NO_3$-N. Furthermore, the initial higher concentrations of phosphorus (500 and 1200 μg/g in IB and OB, respectively) indicated that these materials possibly contained phosphorus functional groups, such as -P=O, -P-O-C, -P=OOH, and P-O-P chains [49]. The nitrogen functional groups and phosphorus functional groups are considered electroactive functional groups that can donate or accept electrons for binding the adsorbates.

The characterization of pine-based biochars derived from Idaho and Orgon sources, using FT-ICR MS with -ESI, indicated that the untreated IB contained compounds with six or fewer oxygens and dominance of $O_4$-$O_6$ classes (extracted by acetone, and therefore, these are less polar/highly aromatic compounds), as shown in Figure 3a. On the other hand, the untreated OB showed a higher content of highly aromatic/more polarizable compounds of the 2-oxygen class, as shown in Figure 3b (Toluene/THF/MeOH extraction). After water treatments, compounds with much higher oxygen numbers were detected in both types of biochar. This observation implied that high oxidizing conditions prevailed during the water treatment. The IB contained a large amount of highly aromatic, oxygen-containing molecules that are possibly alkyl depleted (as shown In Figure 4) as the abundance points were closer to the polycyclic aromatic hydrocarbon limit line. The compounds lying below the polycyclic aromatic hydrocarbon limit line will have planar organic molecules, while compounds lying above the line will have "buckybowl"-type structures (convex−concave pi-conjugated surfaces in hemispherical structures) [50]. On the other hand, the OB showed a limited compositional range in terms of the double bond equivalent (DBE = $c - h/2 + n/2 +1$ for the compound $C_cH_hN_nO_oS_s$) less than 20, and the abundance points lying well below the polycyclic aromatic hydrocarbon limit line. The van Krevelen diagrams (Figure 5) indicated that the contents of aromatic and condensed aromatic compounds were higher in untreated IB samples than in untreated OB samples. The abundance of aromaticity index (AI) increased with the water treatment.

Analysis of the nitrogen and phosphorus adsorption results and correlation of the results with the high-resolution mass spectroscopy data indicated that the adsorption mechanisms operating on the biochar surface are complex. Several mechanisms could synergistically interact, such as physisorption, chemisorption, electrostatic interaction, and ion exchange. Other researchers have reported similar observations [7,9,51]. Dai et al. (2020) investigated the utilization of biochar for nitrogen and phosphorus removal due to the high removal rate and other environmental benefits [52]. Earlier studies achieved a maximum capturing rate of around 0.70 mg/g for ammonium ($NH_4^+$), 40.63 mg/g for nitrate ($NO_3^-$), and 74.47 mg/g for phosphate ($PO_4^{3-}$), using different engineered biochars [7]. Higher adsorption characteristics of IB than OB could be attributed to the high concentrations of alkyl-depleted highly aromatic oxygen-containing molecules. The limited sequestration of phosphorus by IB could be due to the site-specific competition between nitrogen and phosphorus ions. Modification of the biochar with $Ca^{2+}$, $Mg^{2+}$, $Fe^{2+}$, or $Al^{3+}$ cations could have improved the phosphorus sequestration capacity.

## 4. Conclusions

This study focuses on removing water contaminants from the effluent of aquaculture facilities, using woody-based biochars that can reduce eutrophication (oversupplied nutrients) from fish farm effluent. Particularly, this study investigates (a) the nitrogen-ammonia, nitrate-n + nitrite-n, and total phosphorus adsorption rate and amount, as well as saturation time at 3.7–7.5 L/min downstream water flow, (b) the adsorbing and releasing of materials after water treatment, using different pinewood biochars, and (c) the effectiveness level of the proposed water treatment method to remove a significant amount of contaminant, especially nutrients (nitrogen and phosphorus). The results indicate that the proposed approach can improve downstream fish farm waters by adsorbing micronutrients (e.g., nitrogen-ammonia, nitrate-n + nitrite-n, and total phosphorus). Water treatment time and biochar pH are the key parameters that had relatively strong associations with adsorbed compounds. Overall, the results appear more promising as a solution to reduce micronutrients from fish farm effluent. Longer residence times should be applied to identify the optimal treatment time for the adsorption of phosphorus and nitrogen compounds.

Molecular-level characterization of solvent-extracted organics from biochar materials suggests increased levels of highly oxygenated molecules as a function of increasing water-treatment time. Also, the results show the enrichment in organic species with higher molecular weight (higher carbon number) and increased double bond equivalents, with a compositional range similar to that of dissolved organic matter. Upon water treatment, extracted organics revealed higher abundances of compounds with higher H/C and O/C ratios. The engineered biochars can be reused as nutrient-rich soil conditioners or fertilizers after water treatment. This study concluded that the engineered biochars could sequester more nitrogen and phosphorous over more time. Also, the proposed approach can simultaneously increase fish production capacity and support the aquaculture industry in the western U.S. and other regions by improving water quality and enabling aquaculture expansion.

This study has identified the following areas that need further investigation to determine engineered biochar potential to effectively remove nitrogen and phosphorus from fish farm effluents:

- Investigation of different woody-based biochars with greater nutrient removal capacity from aquaculture effluents.
- Investigation of various engineered biochar production with different physicochemical properties to identify a material with higher micronutrient adsorption rates.
- Investigation of modified biochars with other chemical materials, such as magnesium chloride ($Mgcl_2$) or aluminum chloride ($Alcl_3$), to increase nutrient adsorption rate from eutrophic waters.
- Investigation of released materials or pollutants from biochar into downstream receiving waters.

- Investigation of used biochar after water treatment for soil improvement.

**Supplementary Materials:** The following supporting information can be downloaded at https://www.mdpi.com/article/10.3390/pr11102883/s1, Table S1: Biochar results before and after water treatment, Table S2: Water results before and after 0.5 h treatment with IB and OB.

**Author Contributions:** W.F.R.B., E.S., A.M., K.O., M.L.C.-P. and K.S.R. contributed to the conceptualization, methodology, data collection, analyses, and writing the manuscript. W.F.R.B. obtained Oregon biochar, E.S. produced Idaho biochar, and K.O. performed the field experiments. M.L.C.-P., A.M.M. and H.C. performed the characterization of extractable organic compounds 21 T FT-ICR MS. All authors contributed to writing the manuscript. All authors have read and agreed to the published version of the manuscript.

**Funding:** This research was funded by the United States Geological Survey, 104b grant.

**Data Availability Statement:** All FT-ICR MS data files are publicly available via the Open Science Framework (https://osf.io/2yxuk/ (1 October 2022)) at https://doi.org/10.17605/OSF.IO/2YXUK, accessed on 1 October 2022. All data needed to evaluate the conclusions are present in the manuscript and/or the Supplementary Materials.

**Acknowledgments:** The authors wish to acknowledge the United States Geological Survey (USGS) 104b grants for funding this project. The authors also wish to acknowledge inputs, resources, and support from the United States Department of Agriculture, Agricultural Research Service (USDA-ARS), University of Idaho Aquaculture Research Institute (UI-ARI), Riverence Provisions LLC, Idaho Water Resources Research Institute (IWRRI), National High Magnetic Field Laboratory (MagLab) supported by the National Science Foundation Division of Materials Research and Division of Chemistry through DMR 1644779, the State of Florida, and Oregon Biochar Solutions. Publication of this article was funded by the University of Idaho - Open Access Publishing Fund.

**Conflicts of Interest:** The authors declare no conflict of interest.

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
