# Peer review of "Controlling Eutrophication of Aquaculture Production Water Using Biochar: Correlation of Molecular Composition with Adsorption Characteristics as Revealed by FT-ICR Mass Spectrometry"

_processes, doi:10.3390/pr11102883_

Round 1

Reviewer 1 Report

Very nice paper, with a wealth of information, showing how to use sustainable biochar to both fight eutrophication and help farmers enrich land potential. Still, two observations:

1) please, use only SI units

2) you need to remove "the possible adsorption mechanism of nitrate onto unmodified biochar" or move it to the supplementary material, otherwise you need to make a table with "the possible mechanisms” for all other adsorbed components.

Author Response

Reviewer #1:

Very nice paper, with a wealth of information, showing how to use sustainable biochar to both fight eutrophication and help farmers enrich land potential. Still, two observations:

Reply: Thank you so much for your time and comments.

1) please, use only SI units

Reply: The authors have updated the manuscript and used SI units. Please check the highlighted in Yellow.

2) you need to remove "the possible adsorption mechanism of nitrate onto unmodified biochar" or move it to the supplementary material, otherwise you need to make a table with "the possible mechanisms” for all other adsorbed components.

Reply: The authors have updated the manuscript and deleted the "the possible adsorption mechanism of nitrate onto unmodified biochar" under the Discussion section.

Reviewer 2 Report

This manuscript describes a novel and sustainable approach for remediating aquaculture- 26 generated water contamination using various engineered biochars. The language is good. However, there are some major comments of the paper Therefore; I suggest that the authors should have a major revision and collation of the article. For more details see the comments bellow

Q1. In the abstract section “lines 23-25 are confused and need modification.

Q2. Keywords need to be rewritten- short words and arranged alphabetically.

Q3. In introduction section author must mention different materials used in the degradation of adsorption of phosphorous, please use these (J Adv Pharm Edu Res 2018;8(3):59-67) .

Q4. Aim of work must represent in the introduction section.

Q5.  In biochar synthesis. Does biochar used as row or modified with acids like (phosphoric,….etc)?

Q6. Please compare the results between this works with other previous works .

Q7. Please calculate major cations and major anions for water sample before and after treatment.

Q8. The conclusion part is confused and needs modification

Author Response

Reviewer #2:

This manuscript describes a novel and sustainable approach for remediating aquaculture- 26 generated water contamination using various engineered biochars. The language is good. However, there are some major comments of the paper Therefore; I suggest that the authors should have a major revision and collation of the article. For more details see the comments bellow

Q1. In the abstract section “lines 23-25 are confused and need modification.

Reply: Thank you so much for your time and comments. The authors have updated the abstract and moved the unclear sentences to the Introduction section.

Q2. Keywords need to be rewritten- short words and arranged alphabetically.

Reply: The authors have updated the keywords.

Q3. In introduction section author must mention different materials used in the degradation of adsorption of phosphorous, please use these (J Adv Pharm Edu Res 2018;8(3):59-67) .

Reply: Thanks for the comments. The authors believe the mentioned article is not aligned with the study focus and scope of our manuscript. Therefore, we decided to not mention it in our manuscript because if we mention this study, then we have to mention several similar studies.

Q4. Aim of work must represent in the introduction section.

Reply: The authors have provided Subsection “1.3 Study Focus and Objectives” in the Introduction section that explains the details.

Q5.  In biochar synthesis. Does biochar used as row or modified with acids like (phosphoric,….etc)?

Reply: In this study, we used raw biomass feedstocks to produce engineered biochars.  In our future studies, we will modify our biochar with Magnesium chloride (MgCl2) to increase the nitrogen and phosphorus capturing rate.

Q6. Please compare the results between this works with other previous works .

Reply: The authors have added the following sentences to the Discussion section to compare the results of this study with others:

“Other researchers have reported similar observations [7,9,53]. Dai et al. (2020) investigated the utilization of biochar for nitrogen and phosphorus removal due to high removal rate and other environmental benefits [54]. Earlier studies achieved maximum capturing rate around 0.70 mg/g for ammonium (NH4+), 40.63 mg/g for nitrate (NO3), and 74.47 mg/g for phosphate (PO43-), using different engineered biochars [7].”

Q7. Please calculate major cations and major anions for water sample before and after treatment.

Reply: Our team conducted this study in the field a couple of months ago, and currently, we do not have water samples for further analyses. The authors have provided water results in Supplementary Materials, as shown in Table S2 below:

Table S2. Water results before and after 0.5 hr treatment with IB and OB

Water Sample

Nitrate (mg/L)

Total Phosphorus (mg/L)

BT #1

2.26

0.15

BT #2

2.34

0.14

AT IB #1

2.36

0.14

AT IB #2

2.25

0.15

AT IB #3

2.35

0.15

AT OB #1

2.31

0.85

AT OB #2

2.34

0.41

AT OB #3

2.33

0.75

BT: before treatment; AT: after treatment; IB: Idaho biochar; OB: Oregon biochar.

Q8. The conclusion part is confused and needs modification

Reply: The authors have updated the Conclusion section and made several changes to address this comment.

Round 2

Reviewer 2 Report

The manuscript titled “Controlling Eutrophication of Aquaculture Production Water Using Biochar: Correlation of Molecular Composition with Adsorption Characteristics as Revealed By FT-ICR Mass Spectrometry” is suitable for publication in its present form in the journal of processes .